# Characterization and Physiological Differences of Two Primary Cultures of Human Normal and Hypertrophic Scar Dermal Fibroblasts: A Pilot Study

**DOI:** 10.3390/biomedicines12102295

**Published:** 2024-10-10

**Authors:** Natalia M. Yudintceva, Yulia V. Kolesnichenko, Alla N. Shatrova, Nikolay D. Aksenov, Natalia M. Yartseva, Maxim A. Shevtsov, Viacheslav S. Fedorov, Mikhail G. Khotin, Rustam H. Ziganshin, Natalia A. Mikhailova

**Affiliations:** 1Institute of Cytology of the Russian Academy of Sciences, Tikhoretsky 4, 194064 Saint-Petersburg, Russia; koles209@mail.ru (Y.V.K.); shatrova@mail.ru (A.N.S.); aksenovn@gmail.com (N.D.A.); ya.ya-natm27951@yandex.ru (N.M.Y.); shevtsov-max@mail.ru (M.A.S.); fedorovvs.biotech@gmail.com (V.S.F.); khotin@incras.ru (M.G.K.); natalia.mikhailova@incras.ru (N.A.M.); 2School of Medicine and Life Sciences, Far Eastern Federal University, Campus 10 Ajax Bay, Russky Island, 690922 Vladivostok, Russia; 3Shemyakin-Ovchinnikov Institute of Bioorganic Chemistry Russian Academy of Sciences, Miklukho-Maklaya Street 16/10, 117997 Moscow, Russia; rustam.ziganshin@gmail.com

**Keywords:** dermal fibroblast, hypertrophic scar, kapyotype, proliferation, motility, proteome, secretome, soluble and vesicular phase

## Abstract

**Background/Objectives:** Dermal fibroblasts (DFs) are key participants in skin hypertrophic scarring, and their properties are being studied to identify the molecular and cellular mechanisms underlying the pathogenesis of skin scarring. **Methods:** In the present work, we performed a comparative analysis of DFs isolated from normal skin (normal dermal fibroblasts, NDFs), and hypertrophic scar skin (hypertrophic scar fibroblasts, HTSFs). The fibroblasts were karyotyped and phenotyped, and experiments on growth rate, wound healing, and single-cell motility were conducted. **Results:** Comparative analysis revealed a minor karyotype difference between cells. However, HTSFs are characterized by higher proliferation level and motility compared to NDFs. These significant differences may be associated with quantitative and qualitative differences in the cell secretome. A proteomic comparison of NDF and HTSF found that differences were associated with metabolic proteins reflecting physiological differences between the two cells lines. Numerous unique proteins were found only in the vesicular phase of vHTSFs. Some proteins involved in cell proliferation (protein-glutamine gamma-glutamyltransferase K) and cell motility (catenin delta-1), which regulate gene transcription and the activity of Rho family GTPases and downstream cytoskeletal dynamics, were identified. A number of proteins which potentially play a role in fibrosis and inflammation (mucin-5B, CD97, adhesion G protein-coupled receptor E2, antileukoproteinase, protein S100-A8 and S100-A9, protein caspase recruitment domain-containing protein 14) were detected in vHTSFs. **Conclusions:** A comparative analysis of primary cell cultures revealed their various properties, especially in the cell secretome. These proteins may be considered promising target molecules for developing treatment or prevention strategies for pathological skin scarring.

## 1. Introduction

The repair of significant skin damage results in scarring, which disrupts tissue structure and function. The scarring process can be lengthy and may lead to complete regeneration of the skin or pathological repair, resulting in hypertrophic or keloid scar formation [1,2,3]. There are differences between these types of scars, not only clinically but also histopathologically. For example, differences were explored for collagen orientation and bundle thickness [4]. However, the primary diagnostic feature of keloids is their horizontal growth, which supports the distinction between keloids hypertrophic scars, though histopathological differences can be less straightforward [5]. In particular, hypertrophic scars develop within the initial boundaries of the wound and tend to regress over time, while keloid scars grow without restrictions and rarely regress [6]. Currently, potential molecular targets for the differentiation of raised skin scars are being developed. Future targeted studies may provide diagnostic and prognostic markers for keloids and hypertrophic scars [7]. One of the important goals of regenerative medicine is to develop approaches for controlling fibrous tissue formation and the healing process, or to eliminate/reduce pathological scars. Scar treatments include surgical resection, laser and radiation therapy, physiotherapy and medication (injections of triamcinolone). However, any preventive and therapeutic strategies to date, unfortunately, still remain unsatisfactory, creating significant problems in clinical practice and cosmetology [8].

Scarring is a complex, multi-stage process involving numerous cells, cytokines and other factors [9,10,11,12]. However, despite numerous studies, the exact mechanisms underlying scar formation remain unclear. Studies on pathological scaring and its treatment are mainly conducted on small animal models (mice, rats, etc.). Unfortunately, the results of animal studies often differ significantly from clinical studies [13]. First, this is caused by significant differences between the structure of the skin of small experimental animals and humans. In addition, the timing of scarring and the stages of its development for humans are quite long, while experimental animals have relatively short lifespans. The experiments, maintenance and care of large animals are very expensive and inaccessible. An additional problem is the lack of a unified methodology for modeling skin scars (application of linear incisions, puncture or multilayer wounds, long-term non-healing ulcers caused by chemicals, burns, etc.).

An alternative approach for studying the molecular mechanisms of skin repair and scarring involves in vitro experiments using various cell cultures. This approach allows for standardized research conditions and is more accessible to a wide range of researchers. Dermal fibroblasts (DFs) are one of the most important participants in skin repair after damage and scarring. Cell migration during wound healing is activated by various factors such as cytokines, chemokines and growth factors [14,15,16]. Proteins secreted by DFs into the extracellular space (cell secretome) play a key role in the interaction of cells involved in the healing process [17,18,19]. The formation of abnormal scars is due to many factors including severe inflammatory response, poor blood supply, and increased synthesis extracellular matrix (ECM) by fibroblasts which lead to excessive ECM deposition during the process of wound healing. Fibroblasts play an important role in wound healing and scar formation [20] which are essential for ECM production. During the proliferative and remodeling phases, fibroblasts produce large amounts of extracellular components, cytokines, chemokines, growth factors, etc., (cell secretome), which can be conditionally divided into soluble and vesicular phases. Extracellular vesicles (EVs) mediate intercellular communication and are involved in many physiological and pathophysiological processes, including modulation of immune responses, maintenance of homeostasis, inflammation, angiogenesis, and others. EVs obtained from dermal fibroblasts have been shown to contain, in addition to a large set of extracellular matrix components, multiple stimuli for migration, proliferation and inflammatory processes [21]. At the same time, the proteins of soluble phase may have different functional significance and expression levels. Thus, assessing the two phases of the cellular secretome is important as this approach opens new perspectives for identifying potential targets for scar therapy and developing more effective treatments.

However, a huge number of studies have been devoted to the analysis of DF properties isolated from various scar tissue (atrophic, hypertrophic or keloid scar), and analyzed without appropriate control-fibroblasts obtained from a fragment of normal skin from the same site and from the same donor [22,23,24,25]. Despite the simple description of the fibroblast phenotype, different fibroblasts exhibit different gene expression patterns and have different functions [26]. The functional diversity of fibroblasts is due to their embryonic origin, anatomical location in the tissue and microenvironment [27,28]. High fibroblast heterogeneity leads to significant differences in their properties [29,30]. It is known that papillary and reticular fibroblasts make different contributions to the process of repairing damaged tissue [29,31]. Fibroblasts isolated from different areas of the skin [32,33], obtained from patients of different ages [34,35,36,37] and genders [38,39,40] also have a number of differences. The most significant differences are revealed in the cellular secretions of normal and scar fibroblasts [41,42,43].

Studying the molecular and cellular mechanisms underlying pathological scar formation using DFs isolated from normal and hypertrophic scar tissue from the same donor excludes the influence of the aforementioned factors. In the present work, we conducted a comparative analysis of the different properties of fibroblasts isolated from normal skin (NDFs) and hypertrophic scar skin (HTSFs) from the same donor. In particular, we examined karyotype and phenotype, assessed proliferation and cell motility, and analyzed the soluble and vesicular phases of the cell secretome. In addition, the ongoing research is aimed at finding target molecules for developing strategies to treat or prevent pathological scar formation.

## 2. Materials and Methods

### 2.1. Isolation of Dermal Fibroblasts

Skin excision tissue samples with adjacent normal skin were obtained from patient with hypertrophic scar (a 21-year-old male). The scar was formed as a result of a flame burn for about 1.5 years and was localized on the back surface of the hand. Experiments with cells were approved by the Ethics Committee of the Institute of Cytology RAS, protocol no. 20/23 dated 3 October 2023. Cells were isolated from skin fragments using the “explant migration” method. Skin biopsies were washed three times in PBS solution with gentamicin (50 µg/mL), cut into small fragments (1.0 × 1.0 mm), placed on the surface of culture dishes, covered with a coverslip, and cultured in Dulbecco’s Modified Eagle Medium (DMEM, BioinnLabs, Rostov-on-Don, Russia) contained 15% fetal bovine serum (FBS) and gentamicin (Gibco, Carlsbad, CA, USA) in a CO_2_ incubator condition. Cells were passaged upon reaching 80 % confluency using Trypsin-EDTA (0.25%, Gibco, Carlsbad, CA, USA). Further cultivation of DFs was carried out under standard conditions using DMEM with 10% FBS and gentamicin. The study used normal dermal fibroblasts (NDFs) and hypertrophic scar tissue fibroblasts (HTSFs) at passages 6–8.

### 2.2. Karyotype Analysis

Karyotyping of NDFs and HTSFs at the 2nd, 7th and 8th passages was performed. To accumulate cells in the metaphase stage, demecolchicine solution (Sigma-Aldrich, MA, USA) was added to the culture medium at a final concentration of 0.1 µg/mL, and the cells were incubated for 3 h. Hypotonic treatment using a mixture of 0.55% KCl and 1% Na citrate solutions in a ratio of 1:1 for 15–20 min at 37 °C was performed. The cell suspension was fixed using a mixture of methanol and glacial acetic acid in a ratio of 3:1 for 15 min. To prepare metaphase chromosome spreads, the fixed cell suspension was dropped onto dry slides over a water bath at 50–53 °C. Chromosome analysis was carried out using the standard GTG-banding technique for differential staining of chromosomes. The dried chromosome preparations were treated with 0.02% trypsin solution (Difco, NJ, USA) for 2–3 min at 37 °C. The enzyme action was stopped using GKN solution for 15 sec. The chromosomes were then stained with a 2% Giemsa solution (Merck Millipore, MA, USA) for 4 min. Chromosomal structural rearrangements were analyzed automatically using a hardware-software complex, which included an Axio Skop A1 ProgRes MF microscope (Carl Zeiss Microscopy, Jena, Germany), a high-resolution black-and-white CCD camera (Jenoptik, 1360 × 1024 pixels, Jena, Germany), and a computer with the VideoTesT Karyo 3.1 software installed.

### 2.3. Flow Cytometry

Flow cytometry using a CytoFlexS flow cytometer (Beckman Coulter, Brea, CA, USA) equipped with CytExpert software (version 2.0, Brea, CA, USA) was performed.

#### 2.3.1. Phenotype Analysis

For the phenotyping of NDFs and HTSFs, the cell staining buffer (Cell Staining Buffer; BioLegend, San Diego, CA, USA) and a panel of antibodies conjugated with the specific fluorochromes, including PE-conjugated antibodies against CD34 PE (IM1420) Beckman Coulter, Brea, CA, USA), CD73 PE (550257), CD90 PE (561970), CD105 PE (560839), HLA-DR PE (555812) (BD Pharmingen, San Diego, CA, USA), CD44 PE (12-0441-81) eBioscience, San Diego, CA, USA), all from as well as FITC-conjugated antibodies against HLA-ABC FITC (IM1838U) (Beckman Coulter, Brea, CA, USA) and SSEA-3 Alexa Fluor488 (FCMAB141A4) (Merck KGaA, Darmstadt, Germany) were used. Antibodies that do not have specificity to the studied antigens, but correspond to the class and type of antibodies in accordance with the instructions of the manufacturers, were used as isotypic controls (Iso PE (554680) and Iso FITC (555748) BD Pharmingen, San Diego, CA, USA).

#### 2.3.2. Growth Curve

To evaluate the proliferative activity of NDFs and HTSFs, cells were seeded at equal concentrations (2.5 × 10^3^ cell/cm^2^) in 6-well plates (Thermo Fisher Scientific, Waltham, MA, USA). Cell size and autofluorescence levels were assessed using a flow cytometer. Cell counts every 48 h over a 10-day period (240 h) were performed. At each time point cells were harvested and stained with propidium iodide (100 µg/mL; Servicebio, Wuhan Optics Valley, Wuhan, China). Each time point was analyzed three times and in three replications. The proliferative indices (PI) were determined as the ratio of the cell number at the analyzed time point to the cell number of initially seeded. Based on the collected data, the logarithmic growth phase was determined, and a growth curve was plotted for NDFs and HTSFs using GraphPad Prism 10.

### 2.4. Assessment of Cell Motility

To evaluate wound healing speed and single-cell movement, the automated cell imaging system Image ExFluorer (LCI, Namyangju-si, Gyeonggi-do, Republic of Korea) was used.

#### 2.4.1. Wound Healing Assay

To assess wound healing speed, NDFs and HTSFs were cultured in a Culture-Insert 2 Well in µ-Dish 35 mm (Ibidi, Gräfelfing, Germany). Migration assays were performed by seeding the cells into the two-well culture insert. Following cell attachment, insert was delеted, and cell migration was visualized using time-lapse imaging. A fresh DMEM medium supplemented with 10% FBS and gentamicin was applied. Images (resolution of 2560 × 2160 pixels with a pixel size equivalent to 0.67 μm in the x- and y-axes) were obtained using phase contrast illumination and a × 10 objective lens every 15 min for a period of 48 h. The migration area was calculated as follows:Migration area (%) = (A0 − An)/A0 × 100,
where A0 represents the initial scratched area and an represents the remaining wound area at the measurement point.

#### 2.4.2. Single Cell Motility Assay

NDFs and HTSFs were seeded at the same low concentration (2.5 * 10^3^ cell/cm^2^) onto 6-well plates (Thermo Fisher Scientific, Waltham, MA, USA) and allowed to adhere overnight. For visualization and tracking of each cell movement, cell nuclei were stained with 0.1% Hoechst 33342 (Invitrogen, Thermo Fisher Scientific, Waltham, MA, USA). Fluorescence detection was performed at a wavelength of 405 nm using a long-working distance semi-apochromatic lens (S Plan Fluor ELWD) 20× with a numerical aperture of 0.45. Images were captured every 15 min over a 24-h period, with 10 fields of view recorded per well. Images with a resolution of 2560 × 2160 pixels (0.33 μm/pixel), with automatic focusing and real-time focus correction. The images were analyzed using NIS-Elements software with an automatic segmentation module for quantitative assessment and tracking of individual cells. The X-Y coordinates of cells with nuclei larger than 2.88 μm in diameter were identified and recorded. Cell movement tracks were generated based on the nuclear coordinates, filtered for duration (more than 48 h), and the average speed over 24 h was calculated. Experiments were performed three times and in three replications.

### 2.5. Isolation and Separation of the Cell Secretome

Cells were seeded at moderate density in flask (8 × 10^6^ cell/175 cm, Nunc, NY, USA). Once cells reached 70–80% confluence, they were washed five times with serum-free DMEM (Biolot, Russia, Saint-Petersburg, Russia) at 20 min intervals to remove residual serum. Then the washed cells were incubated in serum-free medium DMEM (BioinnLabs, Rostov-on-Don, Russia) for 48 h under CO₂ incubator conditions (5%CO_2_, 37 °C). The secretome was obtained from the conditioned medium (CM) using ultracentrifugation methods [44]. CM was collected into tubes and sequentially cleared of cellular debris by centrifugation at 750× *g* for 10 min at 4 °C, followed by 1500 g for 15 min at 4 °C. To separate soluble and vesicular phases, the supernatant was ultracentrifuged in special tubes at 20,000× *g* for 60 min at 4 °C, and then at 120,000× *g* for 90 min at 4 °C.

The soluble phase was concentrated using 10 kDa JetSpin centrifugal filters (Jet Bio-Filtration Co., Guangzhou, China) by sequential centrifugation at 3000× *g* for 30 min. The vesicular phase was further washed with PBS at 120,000× g for 90 min at 4 °C, the supernatant was removed, and the pellet was resuspended in 200–300 µL PBS. Both fractions were aliquoted and stored at −80 °C. The protein concentration in both phases was measured using the Pierce BCA Protein Assay Kit (Thermo Fisher Scientific, Waltham, MA, USA) according to the manufacturer’s protocol. Measurements were performed at a wavelength of 595 nm using a spectrophotometer (Varioskan LUX, Thermo Fisher Scientific, MA, USA). Calibration curves were used to calculate the extinction coefficient and determine protein concentrations in the secretome samples.

### 2.6. Liquid Chromatography and Mass Spectrometry

LC-MS/MS analysis was made using an Ultimate 3000 Nano LC System (Thermo Fisher Scientific, MA, USA) coupled to the Orbitrap Lumos Tribrid mass spectrometer (Thermo Fisher Scientific) via a nanoelectrospray source (Thermo Fisher Scientific, MA, USA). Samples were loaded on a home-made trap column (50 × 0.1 mm) packed with Reprosil-Pur 200 C18-AQ 5 µm (Dr. Maisch) sorbent in a loading buffer (2% ACN, 98% H_2_O, 0.1% TFA) at a flow rate 4 μL/min. Peptides were eluted from the trap-column on a home-made fused-silica column (300 × 0.1 mm) packed with Reprosil PUR C18AQ 1.9 (Dr. Maisch, Ammerbuch, Germany) into an emitter. The following solutions were used for elution of the peptides: Buffer A—0.1% formic acid; buffer B—80% acetonitrile and 0.1% formic acid. Peptides were eluted with a linear gradient: 3–5% solution B for 5 min 5–35% solution B for 100 min; 35–60% B for 15 min, 60% B during 3 min, 60–99% B for 0.1 min, 99% B during 3 min, 99–3% B for 0.1 min at a flow rate of 500 nl/min. MS1 parameters were as follows: 120 K resolution, 350–1600 scan range, max injection time—auto, AGC target-standard. Ions were isolated using a 1.2 m/z window, preferred peptide match and isotope exclusion. The dynamic exclusion time was set to 20 s. MS2 fragmentation was carried out in HCD mode at 15 K resolution with HCD collision energy 30%, max injection time—80 ms, AGC target—standard. Other settings: charge exclusion—unassigned, 1, >7. For subsequent data analysis, raw MS files were analyzed using Peaks studio 10.0 (Bioinformatics Solutions Inc., Waterloo, Canada).

The list of proteins identified by MS was loaded into the STRING Cytoscape plugin [45,46] with the following settings in order to generate a network of protein-protein interactions (PPI): organism–Homo sapiens; network type–full STRING network; meaning of network edges–evidence; minimum required interaction score–medium confidence (0.4). The resulting network was clustered using the Markov Cluster Algorithm (MCL) algorithm [47] with an inflation parameter set to 4. Using default Cytoscape abilities, PPI underwent group-wise functional enrichment analysis with genome as background and identified clusters as groups. In the resulting networks, several characteristics for nodes and edges were obtained. Node size was set as a degree function. PPIs were color-coded to illustrate 10 major protein clusters with their representative functions or biological process relation. Functional maps were obtained using ClueGO + CluePedia plugin [48] with the following parameters: GO biological process, experimental evidence, and GO term fusion within an interval of 5–8, with pathways showing *p*V < 0.05. All networks were layered via yFiles plugin [yWorks GmbH, Tübingen, Germany]. Venn diagram of intersecting protein groups was obtained via an online tool [https://bioinformatics.psb.ugent.be/webtools/Venn/, accessed on 30 August 2024].

### 2.7. Statistical Analysis

The statistical data for cell motility assay were processed using Office Excel 2016 (Microsoft, Redmond, WA, USA) and GraphPad Prism 10 (GraphPad Software Inc., San Diego, CA, USA). Data are presented as a median with 95% confidence intervals. Normality testing was conducted using the Kolmogorov–Smirnov and Shapiro–Wilk tests. All data were abnormally distributed. When comparing two independent samples of quantitative data, the non-parametric Mann–Whitney U test was used to determine statistically significant differences. Differences were considered statistically significant at *p* < 0.05. The graphs were created using GraphPad Prism 10. Data were tested for normal distribution using the Shapiro–Wilk test [49]. Since all data were abnormally distributed, the non-parametric Mann–Whitney t-test was used to identify significant differences between measurement groups [50,51]. Statistical significance was set at *p* < 0.05.

## 3. Results

### 3.1. Cell Morphology and Karyotype Analysis

Two primary cell cultures were obtained as a series of passages represented by a homogeneous population of dermal fibroblasts as normal dermal fibroblasts (cell line NDF) and hypertrophic scar fibroblasts (cell line HTSF). Cells of both lines were characterized by a bipolar spindle-shaped morphology typical for fibroblasts (Figure 1A). STR analysis confirmed the uniqueness of each cell line and the absence of cross-contamination by other cell lines.

Analysis of the NDFs karyotype at the 7th passage showed that cells with a normal number of 46 chromosomes accounted for 95%, and the proportion of tetraploid cells was 0.6%. The karyotype of the cell population of the line was normal (Figure 1B), and only four cells (4%) with non-clonal structural chromosomal rearrangement (SCR) were identified. Two rearranged chromosomes were found in one of them: chromosome 1 with an internal reorganization of the structure affecting both arms, and chromosome 7 with additional material at the p22 locus of the short arm. Clonal SCR have not been identified (Figure 1B).

The karyotype analysis of the HTSFs was carried out two times (on the 2nd and 8th passages). On the 2nd passage, 96% of the cells of the diploid population had a normal number of 46 chromosomes, the proportion of polyploid cells (tetraploid) was 2.4%. Karyotyping of cells revealed 8% of cells with SCR of which 4% (4 cells) showed the same clonal SCR (translocation of a part of the long arm of chromosome 1 to the long arm of chromosome 12–t (1;12) (q11.2; q24)).

As a result of translocation, three copies of most of the long arm of chromosome 1 were formed, that is, a trisomy along the section of chromosome q11.2-qter (Figure 1B(i)). Repeated study on the 8th passage also did not reveal aneuploidy, i.e., a change in the number of chromosomes in the diploid cell population, which was normal 46 (94%). At the same time, the proportion of tetraploid cells decreased and amounted to 1.6%. The number of SPX also decreased to 2%, clonal SPX–t (1;12) (q11.2; q24) was not detected (Figure 1B(ii)). It is likely that the abnormal structure of chromosomes observed at the 2nd passage arose as a result of the HTSFs adapting to the in vitro environment.

### 3.2. Phenotype Analysis and Curve of Cell Proliferation

Phenotype analysis by flow cytometry revealed similar surface marker expression profile between NDFs and HTSFs. About 90% of both primary cell cultures had markers of mesenchymal stem cells (CD105+, CD90+, CD73+, CD44+), markers of hematopoietic cells such as CD34-, HLA-ABC-, HLA-DR- had nearly 0.1–0,3% primary cell cultures, and the marker of embryonic stem cells (SSEA-3) was obtained for 1.5% cells (Figure 2A, Appendix A). HTSFs show higher proliferative activity compared to NDFs. However, in both cell cultures, a number of active proliferation cells decrease after six days (Figure 3A). Starting from the sixth day of measurement, a separate cell population begins to appear in both primary cell cultures. By the eighth day, the contribution of this cell population to the overall autofluorescence level reaches 10% and 21% for NDFs and HTSFs, respectively, (Figure 2B).

Autofluorescence is often used as a marker of cell aging [52]. Based on the literature data, we suggested that the observed autofluorescence could be caused by lipofuscin [53]. Lipofuscin, which mostly does not decompose, accumulates in aging or slowly dividing cells, whereas proliferating cells decrease its content as they divide [54]. The duration of the experiment without changing of nutrient media and cell passaging probably led to the appearance of a population of aging cells.

### 3.3. Analysis of Cell Motility

The assessment of cell motility is performed using scratch wound healing and the construction of single cell tracks. The wounds healing of NDF and HTSF monolayers (Figure 3A(i)) occurred within 33 and 26 h, respectively. Thus, wounds healing of HTSFs are faster by about 20% compared to NDFs (Figure 3A(ii)). Tracking single cells with rare seeding (Figure 3B(i,ii)) is normalized (Figure 3B(iii)). It has been shown that the average speed of HTSFs is higher compared to the speed of NDFs (Figure 3B(iv)).

### 3.4. Characterization of NDFs and HTSFs Proteome by Mass Spectrometry

To investigate the proteomic profile of NDFs and HTSFs, we performed quantitative liquid chromatography-tandem mass spectrometry (LC–MS/MS) proteomics analyzes on their soluble (sNDFs and sHTSFs) and vesicular (vNDFs and vHTSFs) phases. The number of identified proteins is 2.156 (Appendix A), each group includes about six hundreds of them. The obtained data was shown on a diagram which illustrate common and unique proteins for NDFs and HTSFs (Figure 4A, Appendix A). MCL clustering carried out for four fractions revealed 10 main groups, among which were common groups including extracellular matrix proteins (ECM), proteasome complex, chaperone and specific groups (Table 1). Total protein network and specific groups such as an Actin contraction, Annexin and Calmodulin for vHTSFs are presented on the Figure 4B (Table 1). Another three phase of samples you can see on the Appendix A.

With node-degree mapping, we can see that for both vesicular samples, the same set of proteins played a high role in organizing PPI networks (Figure 4B and Appendix A). GAPDH, ACTB, FN1, HSP90(AA1, AB1) facilitate the majority of shortest protein interactions in observed networks. Interestingly, all of these proteins have been shown to directly bind to each other, and the assembly of this complex may play a significant role in ECM organization. For soluble phase samples, HSP90 contribution for network was lower, as part of interactions were rerouted through albumin.

To reveal possible additional necessary for PPI organization nodes, we plotted a scatter of closeness centrality against degree for all proteins. The scatter function had a linear increase motif (R > 0.4), however, with several outlying nodes. These nodes do not act as hubs for shortest interactions, rather, connect a set of highly interfaced protein neighborhoods [55] (Appendix A). The sets of these nodes were unique for vHTSFs, representing a small protein neighborhood disconnected from main network (Table 2).

No shared function was determined for these unique proteins, as they are not a part of major topological or functional clusters. These are cell periphery and extracellular proteins involved in microtubule organization and membrane trafficking.

Additionally, we analyzed sets of intersecting proteins between all samples, as well as vHTSFs-vNDFs intersection (Appendix A). Between all samples, there were five clusters representing a significant percentage of intersecting proteins: actin cytoskeleton 36%, ECM 27%, histones 9%, keratins 7%, proteasome 5% (Appendix A). Both fibroblast types and fractions shared 319 proteins in total with 5 major clusters: actin cytoskeleton 36%, ECM 27%, histones 9%, keratins 7%, proteasome 5% (See Appendix A for cluster content and calculation). This protein group and its composition characterizes innate role of fibroblasts in tissue regeneration. However, this leaves a number of proteins substantial for proteome- and secretome-based differentiation of fibroblasts.

The intersection between vesicles contained one major network with proteins necessary for GTPase modulation, precisely, RHO, GNA and GABARAP families (Figure 5A). Annotated set of unique vHTSF proteins was observed to be a number of disconnected neighborhoods with varying functions. The unique proteins for vHTSF are presented on Figure 5B.

Interestingly, only this set contained a crucial part of chaperome-HSF1. It has been shown that FN1 expression is regulated with HSF1 upon stress, maintaining tissue integrity [56]. Data of the functional properties of proteins reflected the percentage of proteins in each phase are shown in Figure 6 (data on the number of proteins of each element of the map Appendix A).

Significant quantitative and qualitative differences were identified between groups. The soluble phase of HTSFs contains proteins that regulate leukocyte chemotaxis (18.06%), cell–substrate adhesion (18.06%) and smooth muscle cell–matrix adhesion, negative regulation of wound healing (1.39%) and others, once they could be correlated with inflammation processes and scar formation. Vesicular phase of HTSFs also contains proteins with unique function including proteins with regulation of cell migration (13.51%), negative regulation peptidase activity (17.57%), cell substrate adhesion (12.16%) and others.

## 4. Discussion

Obtaining and characterizing new cell lines from various animal or human tissues, as well as creating cell collections and banks, is an important task for obtaining representative and reproducible scientific results. Studying cell lines isolated from tissues of a single donor makes it possible to exclude the influence of additional factors, such as age and physiological characteristics of donors. In this work, we obtained a pair of unique primary cell cultures of dermal fibroblasts from a single donor (healthy skin and pathological skin scar) and demonstrated their differences in functional characteristics and protein composition of secreted extracellular vesicles.

Fibroblasts in adult skin are in a quiescent state, characterized by minimal cell migration and proliferation. Following injury, cell proliferation and migration significantly are significantly enhanced. The mechanism of hypertrophic scar formation is associated with abnormal proliferation and transformation of fibroblasts in response to damage [57]. There are some key aspects of fibroblast biology, such as cell proliferation, migration and extracellular matrix remodeling, that are often considered together. The analysis of cell secretome allows for the identification and characterization of various proteins that may be involved in the processes of pathologic scar formation and remodeling. This approach could be perspective for identifying potential targets for scar therapy and developing more effective treatment methods.

Separation of cell secretome on soluble and vesicular phases allows for a more detailed assessment of their composition and identification of unique groups and individual proteins of each of the fractions. In this work, we detected significant differences between fibroblasts and protein compositions of their different fractions. However, our focus was mainly directed on the identification and analysis of unique proteins of the vesicular fraction of HTSFs as the most significant in the processes of cellular communication and the development of pathological conditions.

Revealed higher proliferation and motility of HTSFs may be associated with quantitative and qualitative differences in proteins secreted by cells. Vesicular and soluble phases of HTSFs contains proteins with regulation of cell migration (13.51% and 8.33%, respectively), proteins of cell substrate adhesion (12.16%). Some specific proteins involved in cell proliferation (Protein-glutamine gamma-glutamyltransferase K (TGM1) [58] and cell motility (Catenin delta-1 (CTND1)) regulating gene transcription and the activity of Rho family GTPases and downstream cytoskeletal dynamics [59] were detected only for vHTSFs phase.

Our findings regarding proteins in vesicular samples are in concurrence with extracellular vesicle-mediated communication data. According to ExoCarta [ExoCarta.org], more than 50 RAB proteins are known to be secreted via exosomes, including all those observed in fibroblast proteome (RAB5C, RAB7A, RAB30). These are of the key proteins involved in formation and trafficking of exosomes, as well as cargo selecting. Additionally, with VAMP proteins they may be responsible for multivesicular bodies formation [60]. Additionally, as RAB proteins are situated in an interconnected network with RHOA, GABARAP and GNA proteins, fibroblast-derived exosomes have a functional apparatus for vesicle formation, transport and GTP-signaling. Aside from cell-to-cell signaling, same exosomes may directly affect to properties of ECM. In observed samples, several ECM remodeling proteins (integrins, connexins) were found, such as ITGB1 and GJA1 [61]. Recent findings also build upon ECM deposition pathways by adding caveolin to modulating factors. Caveolin is shown to be required for exosomal sorting of ECM protein cargo subsets which favor matrix architecture remodeling [62]. Protein complex of CD9, TSPAN3 and PTGFRN found in exosome may be responsible for regulating filament fusion. Proteins exhibiting ion channel activity (e.g., STOM) could possibly belong to pathways in exosome-driven matrix mineralization [63].

The process of wound healing consists of three stages: hemostasis/inflammation, proliferation, and remodeling. A multitude of strategies, including decrease of inflammation, inhibition of activation and proliferation of fibroblasts, reducing synthesis and deposition of ECM proteins, induce of cell apoptosis and other directed for the prevention of scarring. An abnormal inflammatory response during wound healing results in pathological scarring, such as hypertrophic scars and keloids [64,65] and the searching the targets molecule for its decreasing remains a critical issue during clinical practice.

In the present work, we detected a number unique protein for vHTSFs which can play a role in fibrosis and inflammation. One of them is Mucin-5B (MUC5B) associated with familial interstitial pneumonia and idiopathic pulmonary fibrosis. It was shown, that dysregulated MUC5B expression in the lung may be involved in the pathogenesis of pulmonary fibrosis [66]. It was shown that HTSFs-derived exosomes exhibit a profibrotic property when treated to NDFs, which transformed into a profibrotic phenotype via the SMAD and TAK1 signaling pathways [67].

Number of proteins such as CD97 (ADGRE5), adhesion G protein-coupled receptor E2 (AGRE2), antileukoproteinase (SLPI), protein S100-A8 (S100-A8), protein S100-A9 (S100-A9), caspase recruitment domain-containing protein 14 (CARD14) are involved in inflammatory and cancer process. CD97 takes part in adhesion and signaling processes after leukocyte activation and plays an essential role in leukocyte migration. It was found on most hematopoietic cells and in different cancer (colorectal, gastric, esophageal and pancreatic carcinomas). CD97 also expressed abundantly by smooth muscle cells. This may indicate about a similarity of properties smooth muscle cells with HTSFs obtained from scar. Interestingly, for the adhesion G protein-coupled receptor E2 have been shown interaction with CD55 which implicated in cancer progression [68]. Antileukoproteinase (SLPI) plays a role in regulating the activation of NF-kappa-B and inflammatory responses [69]. This protein is required for normal wound healing, probably by preventing tissue damage by limiting protease activity, that probably can contribute to the formation of scars. Protein S100-A8 and protein S100-A9 have pro-inflammatory activity includes recruitment of leukocytes, promotion of cytokine and chemokine production, and regulation of leukocyte adhesion and migration [70]. They can act as a potent amplifier of inflammation in autoimmunity as well as in cancer development and tumor spread [71]. Caspase recruitment domain-containing protein 14 (CARD14) acts as a scaffolding protein that can activate the inflammatory transcription factor NF-kappa-B and p38/JNK MAP kinase signaling pathways, activates MALT1 proteolytic activity and inflammatory gene expression and also protects cells against apoptosis [72]. Proteomic analysis identified DEFA3 and DEFA1B proteins as unique for vHTSFs. Recent extensive analysis revealed several key roles for defensin proteins [73]. Skin injury promotes expression of defensin, which accelerates wound closure. At the same time, this raised level of expression leads to elevated inflammatory response and fibroblast accumulation in wound area, relying on macrophage recruiting and growth factor production, which also could lead to hypertrophic scarring.

Observed in vesicular fraction and unique for it, DSC3 protein has been shown to interact within a network including desmocolin, PKP, DSG, DSP, and keratin proteins. Dysregulation of these proteins, in fact, leads to blistering during hypertrophic scar formation

Additionally, we detected two specific groups in the vHTSFs phase. The metallothionein group (Metallothionein-1X, -2A, -1G, -1E, -1M) contains a large number of cysteine residues that bind various heavy metals. Transcription of these proteins is regulated by both heavy metals and glucocorticoids. They may be involved in FAM168A anti-apoptotic signaling [74]. Another small group of proteins including protein-glutamine gamma-glutamyltransferase E (TGM3), Cornifin-B (SPRR1B), Small proline-rich protein 2G (SPRR2G) and Loricrin (LOR) are involved in the formation of the cornified envelope (CE), a specialized component consisting of covalent cross-links of proteins beneath the plasma membrane of terminally differentiated keratinocytes. Thus, our results demonstrate a similarity in the protein set between HTSFs and cancer cell, which could be more detail analyzed and considered as a potential target for scar therapy.

Observed in vesicular fraction and unique for it, DSC3 protein has been shown to interact with a neighborhood of desmocolin and such as PKP, DSG, DSP and keratin proteins. Dysregulation of these proteins, in fact, leads to blistering during hypertrophic scar formation [75]. Functional analysis of SPRR1A, a DSC3 interactor, revealed its unique interaction with miRNA during cardiac fibroblast activation upon post-infarction remodeling [76]. Another interactor, TGM1, is a known regulator of fibroblast adhesion in wound healing. Recent findings suggest that this protein also plays a role in distinguishing ECM “gaps” to be filled [77]. TGM protein expression seems to affect wound healing in a strain-dependent manner for mice [78], however, intricate downregulation of this DSC3 protein neighborhood could lead to impaired connective tissue formation and less scarring.

Various therapeutic strategies for scar management and scar formation prevention using genetic and cellular methods are being actively developed. Hypertrophic scars are known to share many common features with tumors including germination into an intact dermis, the absence of a definite boundary between a scar and normal skin, infiltrative growth, long-term stability without spontaneous disappearance and a high recurrence rate after surgery [41]. Hypertrophic scars, as well as tumors, are associated with cell survival, which may be the result of abnormal expression of oncogenes and/or tumor suppressor genes [42]. Tumor-related genes mediate the development of hypertrophic scars by two ways. One of them increases the survival of fibroblasts by mutating tumor suppressor genes (for example, P53, P27, and P16), the second pathway increases the resistance of fibroblasts to apoptosis by expressing oncogenes (for example, c-myc, c-fos, Tenascin-C) [43,44]. The regulatory mechanisms underlying the action of botulinum toxin type A (BTXA) against HS have been shown in Zhang’s work [45]. BTXA reduced the chromosome ten (PTEN) methylation level and downregulated the expression levels of methylation-associated genes. BTXA also inhibited the phosphorylation of phosphoinositide 3-kinase (PI3K) and Akt. Our study did not reveal differences in cell morphology and the phenotypes between NDFs and HTSFs obtained from the same patient. At the same time, the revealed karyotypic features of HTSFs do not exclude the possibility of fibroblasts with this translocation and copying of the long arm of chromosome 1 in the formation of hypertrophied scar tissue.

Currently, it has been shown that fibroblasts have impressive plasticity and heterogeneity, while phenotypically and functionally different subtypes of fibroblasts are involved in various anatomical sites and biological processes [79]. Studies on the heterogeneity of DFs emphasize the role of their different lines in the scaring after injury [30,80]. Fibroblasts undergo significant profibrotic changes with wide shifts in marker expression and over time acquire the mechanically activated phenotype of myofibroblasts in vitro independently of other stimuli, which is a factor limiting their study [81,82]. The most reliable biomimetic models are organotypic cultures using ex vivo intact skin biopsies, which theoretically contain all relevant cell types in their natural organization. However, these models face practical problems such as variability from donor to donor and sample availability [83].

Thus, using DFs from both normal and scar tissues of the same patient controls for various factors, providing a reliable model for studying fibrosis and inflammation mechanisms associated with pathological scar formation. Many key issues, including the full degree of heterogeneity of fibroblasts both in and outside the skin, how the condition changes the fibroblasts (for example, as a result of injury), and how these cells are affected by interactions with other types of skin cells and wounds, have yet to be resolved, but new research methods promise to help.

## 5. Conclusions

In this study, two primary cultures—one from hypertrophic scar tissue (HTSF) and the other from adjacent normal skin (NDF)—were obtained from the same donor. A comparative analysis of these primary cell cultures revealed their various properties, especially in the cell secretome. Several unique vHTSF proteins were identified that may play a role in fibrosis and inflammation. These proteins may be considered as target molecules for developing treatment strategies or preventative measures against pathological scar formation. Further research is required to expand upon these findings, given that this is a pilot study.

## Figures and Tables

**Figure 1 biomedicines-12-02295-f001:**
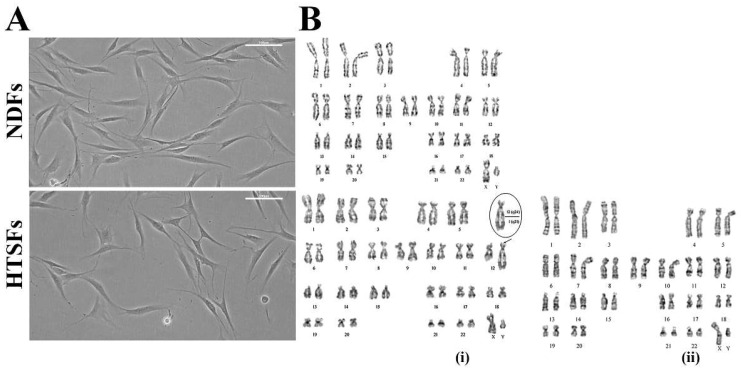
Morphology and karyotype of NDFs and HTSFs. (**A**) Bipolar spindle-shaped morphology of NDFs and HTSFs on the 6th passage. Light microscopy (Nikon Eclipse TS100 (Nikon, Tokyo, Japan). Scale bars, 100 µm. (**B**) NDFs normal karyotype of with 46 chromosomes (top row). HTSFs karyotype on the 2nd (**i**) and 8th passage (**ii**) (bottom row). The arrow indicates the clonal SCR translocation of the long arm of chromosome 1 to the terminal part of chromosome 12. The ellipse indicates the SCR of the chromosome t (1;12) (q11.2; q24), the numbers indicate the regions of chromosomes 1 and 12 in which breaks occurred. HTSFs normal karyotype of with 46 chromosomes (**ii**).

**Figure 2 biomedicines-12-02295-f002:**
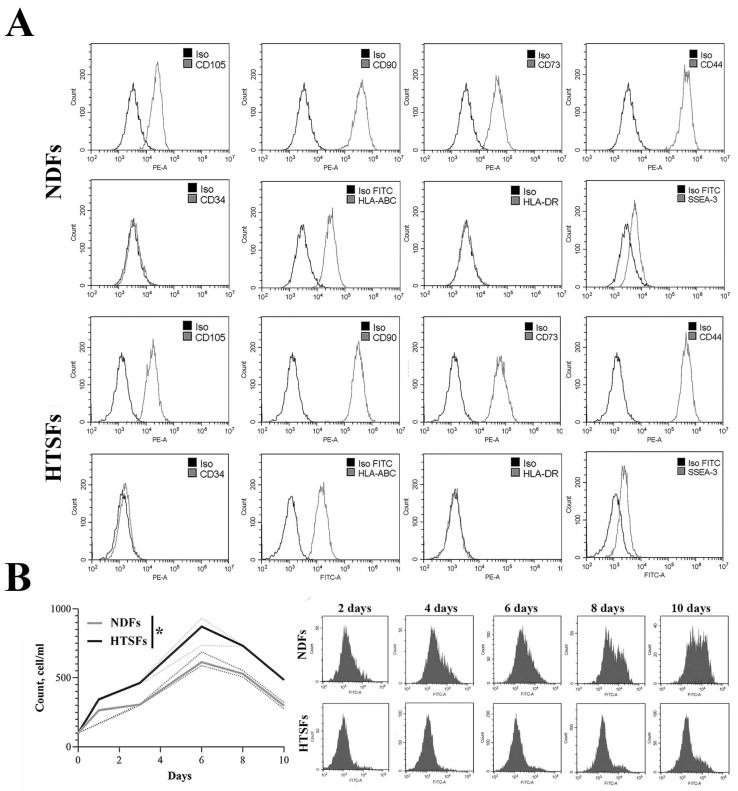
Phenotyping and assessment of cell proliferation. CytoFlexS flow cytometer (Beckman Coulter, Brea, CA, USA). (**A**) Cytograms of surface markers expression (CD105, CD90, CD73, CD44, CD34, HLA-ABC, HLA-DR, SEAA-3). (**B**) Curve of cell proliferation. The average values of the fluorescence intensity are presented. The dashed lines show error bars. The average values are presented, * *p* < 0.05.

**Figure 3 biomedicines-12-02295-f003:**
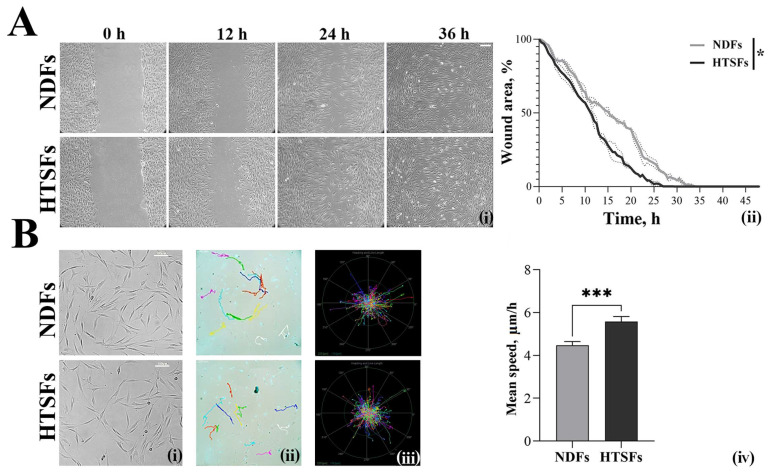
The assessment of cell motility. (**A**) The model of scratch wound (left panel), Scale bar, 100 µm. The wounds healing of cell monolayers during 36 h (**i**). Graphic representation of the rate of wound healing by fibroblasts, right graph (**ii**). The dashed lines show error bars. (**B**) Cells with rare seeding (left panel) (**i**), tracking of single cells (**ii**) and normalized tracking of single cells (**iii**). Diagram showing the differences in speed of single cells (**iv**) (right graph). Scale bars, 100 µm. The average values are presented, **p* < 0.05, *** *p* < 0.001.

**Figure 4 biomedicines-12-02295-f004:**
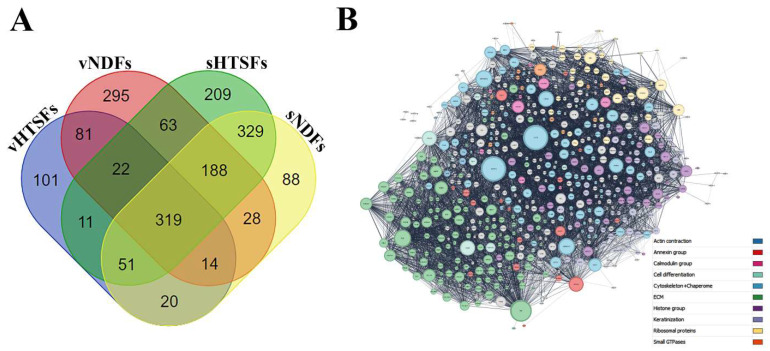
Proteome of the NDF and HTSF cells. (**A**) Venn diagram of the number identified proteins shared between the NDF and HTSF secretomes of soluble and vesicular fractions. (**B**) Size and color-coded interaction network for vHTSFs sample.

**Figure 5 biomedicines-12-02295-f005:**
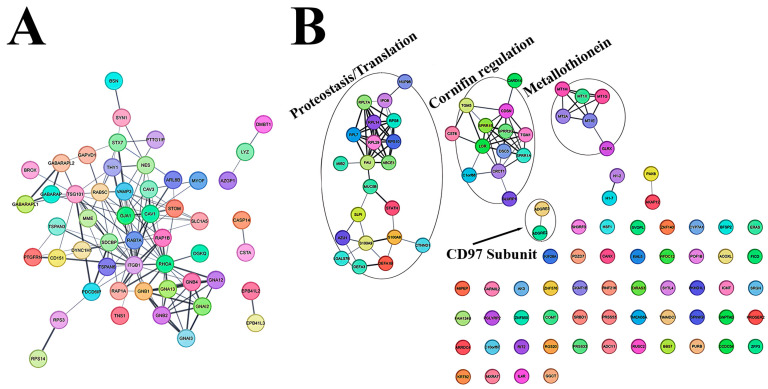
Detailed STRING networks. (**A**) A network of protein intersection for vHTSF and vNDF samples. (**B**) Interaction networks of unique vHTSFs proteins presented as a disorganized set.

**Figure 6 biomedicines-12-02295-f006:**
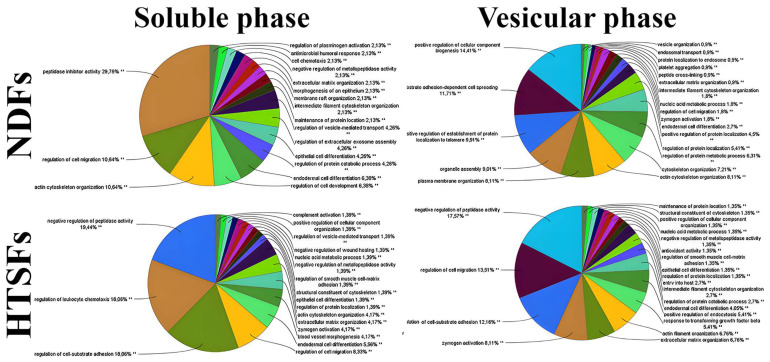
GO functional cluster pie charts.

**Table 1 biomedicines-12-02295-t001:** Major protein clusters.

	Soluble Fraction	Vesicular Fraction
NDFs	Cell differentiation, Chaperome, ECM, Hemostasis, Histone group, mRNA binding, Vesicular transport, Proteasomal proteins, Keratinization, *Lysosomal proteins*	Chaperome, Cytoskeleton, ECM, Ribosomal proteins, Histone group, Small GTPases, Chaperome, *Vesicular proteins group*, Vesicular transport, *tRNAgroup*
HTSFs	Cell differentiation, Chaperome, ECM, Hemostasis, Histone group, mRNA binding, Vesicular transport, Proteasomal proteins, Cytoskeleton, *Carbon metabolism*	Cytoskeleton, Chaperome, ECM, Ribosomal proteins, Histone group, Small GTPases, Cell differentiation, Keratinization, *Actin contraction*, *Annexin group*, *Calmodulin group*

Note: The italic highlights clusters of proteins unique to each group.

**Table 2 biomedicines-12-02295-t002:** Identified high centrality nodes.

vHTSFs	vNDFs	sHTSFs	sNDFs
*STARD9*	NBPF10	NBPF10	NBPF10
*PRSS53*	NBPF19-2	NBPF19-2	NBPF19-2
*TRAPPC2L*	*SLC43A3*		
*ARRDC4*	*FAM234A*		

Note. The italic highlights clusters of proteins unique to each group.

## Data Availability

The datasets used and/or analyzed during the current study are available from the corresponding author upon reasonable request.

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
