# Peer review of "Characterization and Physiological Differences of Two Primary Cultures of Human Normal and Hypertrophic Scar Dermal Fibroblasts: A Pilot Study"

_biomedicines, 2024, doi:10.3390/biomedicines12102295_

Round 1

Reviewer 1 Report

Comments and Suggestions for Authors

The manuscript presents results that compare two cell lines: normal and hypertrophic scar dermal fibroblasts. The biggest problem of this study is size of the group - material collected from 1 patient - it is as if without any biological repetition. Such results can be published at most as a curiosity, preliminary results, short report, but not original article. Apart from that, interesting results, well-described methodology, and statistics performed within a single individual.

Minor:

Line 102 – it should by CO2

Author Response

Reviewer Comments:

Reviewer 1

Major

Comment 1: The manuscript presents results that compare two cell lines: normal and hypertrophic scar dermal fibroblasts. The biggest problem of this study is size of the group - material collected from 1 patient - it is as if without any biological repetition. Such results can be published at most as a curiosity, preliminary results, short report, but not original article. Apart from that, interesting results, well-described methodology, and statistics performed within a single individual.

Answer 1:

Thank you very much for your comment! We understand that the problem of the study is the presented data on one patient (a scar on his skin and healthy tissue). It is very difficult to obtain paired biological material of this kind. However, we considered that an in-depth analysis of cell culture pair obtained even from one patient gives scientists a lot of information and guidelines for planning a large-scale study on a large sample of patients in the future. At the same time, the advantage of our study can be considered the absence of individual genetic differences in the patient's tissue cells, age, gender differences and others, thus the experiment excludes patient-specific physiological differences in the material, which are always present when analyzing a large, but random sample set.

We will agree if it is possible to publish our data as a preliminary result or a short report, if such an opportunity exists.

Minor

Comment 1: Line 102 – it should by CO2

Answer 1: It was corrected (Line 118).

Reviewer 2 Report

Comments and Suggestions for Authors

In the paper “Characterization and physiological differences of two cell lines of human normal and hypertrophic scar dermal fibroblasts,” the authors characterized and compared two skin fibroblast cell lines isolated from the same donor, one healthy and one with a scar. I find this article interesting. However, I have the following comments:

Major:

  1. It is suggested that the authors select at least one protein marker from the mass spectrometry analysis and validate and compare its expression in two cell lines by western blot.
  2. Authors are suggested to stimulate cells with selective growth factors and observe their effect in a cell motility study as a control.

Minor:

  1. The legends in the Figures should be legible. The font size should be the same.

Comments on the Quality of English Language

 Minor editing of English language required.

Author Response

Reviewer 2

Major

Comment 1: It is suggested that the authors select at least one protein marker from the mass spectrometry analysis and validate and compare its expression in two cell lines by western blot.

Answer 1: In the present work, we have detected all the common proteins of the cellular secretome, including extracellular matrix proteins (ECM), proteasome complex, chaperones etc, and, in particular, some unique proteins of the vesicular fraction of vHTSFs, which can be considered as promising target molecules. We plan to obtain a comparative assessment of the expression of some group proteins (e.g., extracellular matrix proteins (fibronectins, collagens etc.) and chaperones (Hsp70, Hsp90 ect.)) from primary cultures of paired dermal fibroblasts by Western blotting as a next step of our study from at least three independent pairs of scar and normal fibroblasts.

Comment 2: Authors are suggested to stimulate cells with selective growth factors and observe their effect in a cell motility study as a control.

Answer 2: Thanks for your comment! It is well known that lysophosphatidic acid (LPA) is involved in both normal wound healing and pathological fibrosis. LPA enhances fibroblast proliferation, migration and contraction, and induces the expression of pro-fibrotic mediators such as connective tissue growth factor. We have stimulated cell motility using LPA as a control (these data are not included in the manuscript). The results obtained did not reveal a significant effect of LPA on cell motility (Figure 1, A, B).

Figure 1. Graphic representation of the rate of wound healing by fibroblasts. *p <0.05, ** p <0.01, *** p <0.001, **** p <0.0001.

Minor

Comment 1: The legends in the Figures should be legible. The font size should be the same.

Answer 1: Figures 2, 3, 4, 5 and 6 were corrected.  

Reviewer 3 Report

Comments and Suggestions for Authors

Dear Editors,

The study titled "Characterization and physiological differences of two cell lines of human normal and hypertrophic scar dermal fibroblasts" provides a comparative analysis of fibroblasts derived from normal skin (NDFs) and hypertrophic scar tissue (HTSFs) from the same donor. The research aims to uncover key molecular and cellular differences that contribute to the pathogenesis of hypertrophic scars. The authors used a range of techniques including karyotyping, proteomic analysis, and assays for proliferation and motility, to identify significant distinctions between NDFs and HTSFs, particularly in their secretome composition and cellular behavior. While the findings offer valuable insights into potential therapeutic targets for hypertrophic scarring, the study has limitations, such as a small sample size and lack of functional validation for the identified proteins.

Major

- The title needs to be changed as it does not reflect the cell type used in this study. The authors have isolated primary cells and did not use any cell lines.

- The manuscript needs extensive English language editing as to the use of grammar and orthograph.

- How did the authors distinguish hypertrophic scars from keloids in their study, given the distinct clinical and pathological features between the two types of scarring?

- One of the major drawbacks of the study is the use of fibroblasts derived from a single donor. While this controls for inter-individual variability, the small sample size limits the generalizability of the findings. The authors themselves acknowledge the need to study more donor samples to confirm the observed differences. Additionally, age-related factors in scarring may play a significant role, which cannot be addressed with a single 21-year-old donor. It is highly advisable to perform experiments on additional patients’ cells.

- The study’s conclusion remain merely descriptive and lacks any substantial functional data. Many proteins were identified as potential contributors to hypertrophic scarring, particularly in the vesicular phase of HTSFs, yet no functional assays were conducted to validate their involvement in scar formation or inflammation. The phenotypic effects of a knockdown/inhibition or an overexpression/activation of at least one of the identified proteins would strengthen these claims.

- The discussion on how the karyotypic abnormalities may contribute to hypertrophic scarring or cell behavior is insufficient.

- The physiological role of the vesicular vs the soluble phase of the cell secretome should be mentioned in the introduction.

- There is an extensive use of literature reviews, particularly within the introduction. The authors are advised to refer to original papers instead.

- The conclusions section needs to be rewritten to better reflect the findings.

Minor

- Is there a particular reason for why the authors studied the cells at passage 6 to 8?

- The individual references of the used antibodies have to be included in the methods.

- In line 243, the Student’s test is not used to check for normality.

- The Supplementary materials are nowhere to be found.

Comments on the Quality of English Language

The paper's English needs extensive editing.

Author Response

Reviewer 3.

Major

Comment 1: The title needs to be changed as it does not reflect the cell type used in this study. The authors have isolated primary cells and did not use any cell lines.

Answer 1: The title was changed on “Characterization and physiological differences of two primary cultures of human normal and hypertrophic scar dermal fibroblasts

Comment 2: The manuscript needs extensive English language editing as to the use of grammar and orthograph.

Answer 2: We would like to send our manuscript for extensive English language editing.

Comment 3: How did the authors distinguish hypertrophic scars from keloids in their study, given the distinct clinical and pathological features between the two types of scarring?

Answer 3: Skin fragments were obtained from the clinic from a patient with the diagnosis (Figure 1).

Figure 1. The scars deformation of the back surface of the hand following a flame burn. The depth of skin damage is mixed type (second and third degrees). The duration of hypertrophic scar formation is about 1.5 years. The hypertrophic scar area is detected by black color and the adjacent area of normal skin is indicated by a red ellipse.

Comment 4: One of the major drawbacks of the study is the use of fibroblasts derived from a single donor. While this controls for inter-individual variability, the small sample size limits the generalizability of the findings. The authors themselves acknowledge the need to study more donor samples to confirm the observed differences. Additionally, age-related factors in scarring may play a significant role, which cannot be addressed with a single 21-year-old donor. It is highly advisable to perform experiments on additional patients’ cells.

Answer 4:

Thank you very much for your comment! We understand that the problem of the study is the presented data on one patient (a scar on his skin and healthy tissue). It is very difficult to obtain paired biological material of this kind. However, we considered that an in-depth analysis of cell culture pair obtained even from one patient gives scientists a lot of information and guidelines for planning a large-scale study on a large sample of patients in the future. At the same time, the advantage of our study can be considered the absence of individual genetic differences in the patient's tissue cells, age, gender differences and others, thus the experiment excludes patient-specific physiological differences in the material, which are always present when analyzing a large, but random sample set. Currently we are conducting a similar analysis of several primary cultures of DFs obtained from other patients with similar data (age, scar type and localization).

Comment 5: The study’s conclusion remains merely descriptive and lacks any substantial functional data. Many proteins were identified as potential contributors to hypertrophic scarring, particularly in the vesicular phase of HTSFs, yet no functional assays were conducted to validate their involvement in scar formation or inflammation. The phenotypic effects of a knockdown/inhibition or an overexpression/activation of at least one of the identified proteins would strengthen these claims.

Answer 5: Thank you very much for your comments and suggestions! Our proteomic analysis identified DEFA3 and DEFA1B proteins as unique for vHTSFs. Recent extensive analysis revealed several key roles for defensin poteins [74]. Skin injury promotes expression of defensin, which accelerates wound closure. At the same time, this raised level of expression leads to elevated inflammatory response and fibroblast accumulation in wound area, relying on macrophage recruiting and growth factor production, which also could lead to hypertrophic scarring.

Observed in vesicular fraction and unique for it, DSC3 protein has been shown to interact with a neighborhood of desmocolin and such as PKP, DSG, DSP and keratin proteins. Disregulation of these proteins, in fact, leads to blistering during hypertrophic scar formation [76]. Functional analysis of SPRR1A, a DSC3 interactor, revealed it’s unique interaction with miRNA during cardiac fibroblast activation upon post-infarction remodeling [77]. Another interactor, TGM1, is a known regulator of fibroblast adhesion in wound healing. Recent findings suggest that this protein also plays a role in distinguishing ECM “gaps” to be filled [78]. TGM protein expreession seems to affect wound healing in a strain-dependend manner for mice [79], but complex down-regulation of this DSC3 protein neighborhood may result in impaired connective tissue formation and less scarring. We have not yet performed functional assays to confirm the involvement of these proteins in scarring or inflammation, but such studies are planned as a next step in our research.

Comment 6: The discussion on how the karyotypic abnormalities may contribute to hypertrophic scarring or cell behavior is insufficient.

Answer 6. In humans, genetic abnormalities may arise in tissue cells that lead to adverse effects (the most well-known example being cancer cells and tumor development, but many others). We cannot at this time interpret the significance of the genetic rearrangements we found in the karyotype of scar tissue fibroblasts on their properties and characteristics. However, we believe it is important to publish the fact of chromosomal rearrangements, as this information is also important for other researchers working in a similar field.

Comment 7: The physiological role of the vesicular vs the soluble phase of the cell secretome should be mentioned in the introduction.

Answer 7:  Additional information added to the Introduction.

During the proliferative and remodeling phases, fibroblasts produce large amounts of extracellular components, cytokines, chemokines, growth factors etc. (cell secretome), which can be conditionally divided into soluble and vesicular phases. Extracellular vesicles (EVs) mediate intercellular communication and are involved in many physiological and pathophysiological processes, including modulation of immune responses, maintenance of homeostasis, inflammation, angiogenesis, and others. EVs obtained from dermal fibroblasts have been shown to contain, in addition to a large set of extracellular matrix components, multiple stimuli for migration, proliferation and inflammatory processes [17]. At the same time, the proteins of soluble phase may have different functional significance and expression levels. Thus, assessing the two phases of the cellular secretome is important as this approach opens new perspectives for identifying potential targets for scar therapy and developing more effective treatments.

Comment 8: There is an extensive use of literature reviews, particularly within the introduction. The authors are advised to refer to original papers instead.

Answer 8: Original articles were added to the introduction. [4,5,7,11,12,15,16, 18,19,36,37,40,42,43].

Comment 9: The conclusions section needs to be rewritten to better reflect the findings.

Answer 9: The conclusions have been edited. Two primary cultures of dermal fibroblasts from hypertrophic scar (HTSF) and adjacent normal skin (NDF) were obtained from the same donor. A comparative analysis of primary cell cultures revealed their various properties, especially in cell secretome. A number of unique vHTSF proteins have been identified that may play a role in fibrosis and inflammation and may be considered as target molecules for developing treatment strategies or preventing the formation of pathological scars. Further research is required.

Minor

Comment 1: Is there a particular reason for why the authors studied the cells at passage 6 to 8?

Answer 1: Yes, this work required a large number of cells, especially to obtain cellular secretomes. For this purpose, cells at passage from 6 to 8 were used.

Comment 2: The individual references of the used antibodies have to be included in the methods.

Answer 2:  It was corrected.

Comment 3: In line 243, the Student’s test is not used to check for normality.

Answer 3: It was corrected.

Comment 4: The Supplementary materials are nowhere to be found.

Answer 4: Supplementary materials were added to the site along with the manuscript when it was uploaded.

Round 2

Reviewer 1 Report

Comments and Suggestions for Authors

the manuscript has been corrected

Author Response

Thanks for your comment and suggestions! 

Reviewer 3 Report

Comments and Suggestions for Authors

While the authors have addressed many of the reviewers' comments, it remains clear that additional functional data and increasing the number of patient samples would significantly strengthen the findings. Given the use of fibroblasts from a single donor, suggesting the inclusion of "a pilot study" in the title may be appropriate, as it reflects the preliminary nature of the research and helps manage expectations regarding its generalizability. This needs to be clearly shown in the limitations within the discussion.

Comments on the Quality of English Language

English editing is required

Author Response

Reviewer

Comments and Suggestions for Authors: While the authors have addressed many of the reviewers' comments, it remains clear that additional functional data and increasing the number of patient samples would significantly strengthen the findings. Given the use of fibroblasts from a single donor, suggesting the inclusion of "a pilot study" in the title may be appropriate, as it reflects the preliminary nature of the research and helps manage expectations regarding its generalizability. This needs to be clearly shown in the limitations within the discussion.

Answer: Thanks for your comment and suggestions! The title was changed on “Characterization and physiological differences of two primary cell cultures of human normal and hypertrophic scar dermal fibroblasts: a pilot study”. The conclusions section was corrected.
